# Assessing the potential of free tropospheric water vapour isotopologue satellite observations for improving the analyses of convective events

Matthias Schneider[1], Kinya Toride[2,3,a,b], Farahnaz Khosrawi[1,c], Frank Hase[1], Benjamin Ertl[1,4], Christopher J. Diekmann[1,d], and Kei Yoshimura[2,5]

[1]Institute of Meteorology and Climate Research (IMK-ASF), Karlsruhe Institute of Technology, Karlsruhe, Germany
[2]Institute for Industrial Science, The University of Tokyo, Chiba, Japan
[3]Department of Atmospheric Sciences, University of Washington, Seattle, WA, USA
[4]Steinbuch Centre for Computing (SCC), Karlsruhe Institute of Technology, Karlsruhe, Germany
[5]Earth Observation Research Center, Japan Aerospace Exploration Agency, Japan
[a]now at: Cooperative Institute for Research in Environmental Sciences, University of Colorado Boulder, Boulder, Colorado, USA
[b]now at: NOAA Physical Sciences Laboratory, Boulder, Colorado, USA
[c]now at: Jülich Supercomputing Centre (JSC), Forschungszentrum Jülich GmbH, Jülich, Germany
[d]now at: Telespazio Germany GmbH, Darmstadt, Germany

**Correspondence:** M. Schneider
(matthias.schneider@kit.edu)

**Abstract.** Satellite-based observations of free tropospheric water vapour isotopologue ratios (HDO/$H_2O$, expressed in form of the $\delta$-value $\delta D$) with good global and temporal coverage have become recently available. We investigate the potential of these observations for constraining the uncertainties of the atmospheric analyses fields of specific humidity ($q$), temperature ($T$), and $\delta D$ and of variables that capture important properties of the atmospheric water cycle, namely the vertical velocity ($\omega$), the latent heating rate ($Q_2$), and the precipitation rate (Prcp). Our focus is on the impact of the $\delta D$ observations relative to the impact achieved by the observation of $q$ and $T$, which are much easier to be observed by satellites and routinely in use for atmospheric analyses. For our investigations we use an Observing System Simulation Experiment, i.e. simulate the satellite observations of $q$, $T$, and $\delta D$ with known uncertainties and coverage (e.g. observations are not available for cloudy conditions, i.e. at locations where the atmosphere is vertically unstable). Then we use the simulated observations within a Kalman filter based assimilation framework in order to evaluate their potential for improving the quality of atmospheric analyses. The study is made for low latitudes (30°S to 30°N) and for 40 days between mid-July and end of August 2016. We find that $q$ observations have generally the largest impacts on the analyses quality, and that $T$ observations have overall stronger impacts than $\delta D$ observations. We show that there is no significant impact of $\delta D$ observations for stable atmospheric conditions; however, for very unstable conditions the impact of $\delta D$ observations is significant and even slightly stronger than the respective impact of $T$ observations. The very unstable conditions are rare, but related to extreme events (e.g. storm, flooding), i.e. the $\delta D$ observations significantly impact on the analyses quality of the events that have the largest societal consequences. The fact that no satellite observations are available at the location and time of the unstable conditions indicates a remote impact of $\delta D$ observations that are available

elsewhere. Concerning real world applications, we conclude that the situation of $\delta$D satellite observations is very promising, but that further improving the model's linkage between convective processes and the larger scale $\delta$D fields might be needed for optimising the assimilation impact of real world $\delta$D observations.

## 1 Introduction

Clouds and water vapour control atmospheric radiative heating/cooling and condensation or evaporation of water determine where latent heat is released or consumed. The heating patterns then drive the atmospheric circulation, whereby in particular vertical transport causes additional evaporation/condensation and impacts the distribution of water vapour and clouds, which in turn again modifies the latent and radiative heating patterns of the atmosphere. This strong coupling between moisture pathways, diabatic heating and atmospheric circulation is responsible for important climate feedback mechanisms (e.g. Sherwood et al., 2014; Bony et al., 2015) and is often connected to the evolution of severe weather events (e.g. Fink et al., 2012; Evans et al., 2017). In this context, it is rather worrisome that the diabatic heating rates and the related vertical transport obtained from different current global reanalyses show significant inconsistencies (e.g. Chan and Nigam, 2009; Ling and Zhang, 2013).

For the generation of daily and global scale analyses, the operational assimilation systems assimilate the outgoing microwave or infrared radiation (e.g. Eyre et al., 2022). This radiation contains information on the atmospheric state (mostly atmospheric specific humidity, $q$, and temperature, $T$). There are many different satellites that measure this radiation spectrally resolved, including operational weather satellites, like the European Meteosat- and Metop-series (https://www.eumetsat.int/our-satellites/meteosat-series and https://www.eumetsat.int/our-satellites/metop-series, respectively).

In this study, we investigate the information that free tropospheric $\delta$D observations can offer in addition to the information provided by the observations of $q$ and $T$ for improving the analyses.

The $\delta$D value is calculated as the ratio between the D and H isotopes in water vapour relative to a standard ratio

$$\delta\text{D} = \frac{\text{HDO}/\text{H}_2\text{O}}{R_{\text{VSMOW}}} - 1, \tag{1}$$

where $\text{H}_2\text{O}$ and HDO are the concentrations of all the isotopologues containing two H isotopes and one H and one D isotope, respectively. The Vienna Standard Mean Ocean Water ratio of the two isotopologues ($R_{\text{VSMOW}} = 3.1152 \times 10^{-4}$) is a standard ratio typically encountered in ocean water. The reason why $\delta$D is of particular interest is: firstly, it can be observed with a reasonable precision by satellite on a daily and almost global scale (e.g. Diekmann et al., 2021a) and secondly, HDO enrichment or depletion contains information on vertical transport and convective processes. For vertical or horizontal mixing between dry/depleted and humid/enriched water masses, HDO tends to be enriched (Noone et al., 2011; González et al., 2016). On the contrary, recurring evaporation and condensation in the context of convective activity causes a strong HDO depletion (e.g. Bony et al., 2008; Risi et al., 2008; Noone, 2012; Field et al., 2014; Galewsky et al., 2016; Diekmann et al., 2021b).

During the last 15 years, tropospheric $\delta$D products have been developed for different satellite sensors (e.g. Worden et al., 2007; Frankenberg et al., 2009; Schneider and Hase, 2011; Lacour et al., 2012; Boesch et al., 2013; Worden et al., 2019; Schneider et al., 2020). Meanwhile, different weather and climate models have the water isotopologues and the relevant physical processes implemented and can provide modelled isotopologue fields on a global and regional scale at different horizontal

resolutions (e.g. Yoshimura et al., 2008; Risi et al., 2010; Werner et al., 2011; Pfahl et al., 2012; Eckstein et al., 2018; Tanoue et al., 2023). This offers advanced opportunities for studying atmospheric moisture processes with water isotopologues.

The tropospheric water vapour isotopologue composition has been used for investigating water cycle related biases in atmospheric models (e.g. Risi et al., 2012; Field et al., 2014; Schneider et al., 2017), processes involving clouds or precipitation (e.g. Webster and Heymsfield, 2003; Worden et al., 2007; Blossey et al., 2010; Field et al., 2010; Bailey et al., 2015; Diekmann et al., 2021b), local diurnal-scale moisture transport (Noone et al., 2011; González et al., 2016), and large-scale moisture transport (e.g. Noone, 2012; González et al., 2016; Lacour et al., 2017; Dahinden et al., 2021).

We use a data assimilation framework together with an OSSE (Observation System Simulation Experiment) to document the added value of the free tropospheric $\delta$D satellite observations, i.e. we simulate satellite observations and then evaluate the theoretical impact of assimilating the observations. This assimilation framework was presented in Yoshimura et al. (2014) and has already been applied by Toride et al. (2021) and Tada et al. (2021). Here we simulate the observations in line with the temporal and horizontal coverage achieved by the IASI (Infrared Atmospheric Sounding Interferometer, Clerbaux et al., 2009) satellite sensor. We simulate the IASI data of $q$, $T$, and $\delta$D as generated for the free troposphere by using the retrieval processor MUSICA (MUlti-platform remote Sensing of Isotopologues for investigating the Cycle of Atmospheric water, Schneider et al., 2016, 2022). We evaluate the analyses of the atmospheric fields of $q$, $T$, $\delta$D, the vertical velocity ($\omega$), the latent heating rate ($Q_2$), and the precipitation rate (Prcp). The latter three are strongly coupled and linked to climate feedbacks and weather events. The atmospheric dynamics (expressed among others by $\omega$) is coupled to $Q_2$, which in turn affects the vertical thermal structure and thus dynamics. Prcp describes the removal of moisture from the atmosphere, which in turn affects the $Q_2$ and radiative heating potential.

This study is complementary to Toride et al. (2021), where observations from different platforms and different temporal and spatial coverages were used (satellite, radiosonde and surface observations). The different observational techniques provide diverse information; however, using observations that have a spatial and temporal coverage that differs from the coverage of the IASI $\delta$D data, makes it difficult to understand whether an improvement in the analysis is due to the complementarity of the information provided by $\delta$D or from the complementary coverage (the coverage of IASI $\delta$D is much better than the coverage of radiosonde data and more homogeneous than the coverage of the surface data and data from geostationary satellites, see Figs. S4 and S5 in the Supporting Information of Toride et al., 2021). In our OSSE all observations have the same spatial and temporal coverage (the coverage of the MUSICA IASI water isotopologue satellite data, Diekmann et al., 2021a), which assures that any improvement in the analysis by an additional assimilation of $\delta$D is due to the complementarity of the information provided by $\delta$D and not affected by different coverages. Furthermore, we investigate the assimilation of $\delta$D in addition to the assimilation of satellite observations of $q$ and $T$. The latter (i.e. IASI observations of $T$) were not considered in Toride et al. (2021), despite the fact that they are available with good quality. Moreover, in addition to the general impact study given by Toride et al. (2021), this work investigates the situations when the isotopologue observations can make a unique contribution (versus the situations when they have no significant impact).

In Tada et al. (2021) real IASI $\delta$D observations (only $\delta$D observations) were assimilated and it was shown that such assimilation leads to a better agreement with the ERA5 reanalyses (Hersbach et al., 2020) than not assimilating any data. However,

they did not investigate the much larger impact that can already be achieved by assimilating more easily observable data like $q$ and $T$. In this context, our study has a very different focus: we use the assimilation of the easily observable data ($q$ and $T$) as the reference and evaluate the impact of additionally assimilating $\delta$D observations.

The manuscript is structured as follows: section 2 describes the simulated data and the OSSE, the performed assimilation experiments, and the analysed atmospheric variables and the methods used for evaluating their quality. In Sect. 3, we give an overview on the analyses quality improvements achieved by the different assimilation experiments. Section 4 examines for what atmospheric conditions the $\delta$D observations have the strongest impact on the analyses and it briefly discusses some challenges that have to be overcome for achieving an optimal $\delta$D assimilation impact for real world analyses. A summary of the study is given in Sect. 5.

## 2 Data and evaluation

### 2.1 Data simulations

We use the isotopologue enabled atmospheric general circulation model IsoGSM (Yoshimura et al., 2008) and simulate the atmospheric state for the two months of July and August 2016, in 6 hour time steps, with a spectral model grid resolution T62 (about 200 km horizontal resolution and 28 vertical sigma levels). We use this simulation as the truth and refer to it in the following as the nature data ($x_{n_{i,j}}$, where the index $i$ indicates the time step and the index $j$ the location).

For our OSSE we consider that a thermal infrared sensor like IASI offers no free tropospheric trace gas products in the presence of mid- and high-level clouds, so we limit the observational data availability to time steps and locations where the model is free of mid- and high-level clouds. Furthermore, we take into account IASI's high horizontal resolution (ground pixel diameter of about 12 km at nadir), which is much finer than the 200 km horizontal resolution of IsoGSM. Typically there are about 10-20 high quality MUSICA IASI observations each 12 hours in the $200 \times 200$ km area, that is represented by IsoGSM (Diekmann et al., 2021a). We simulate MUSICA IASI observations of $q$, $T$, and $\delta$D in the middle troposphere (at about 550 hPa) and consider the different horizontal representations of model and observation when setting up the observational error variance ($\sigma_o^2$). For this purpose, we estimate $\sigma_o^2$ as the sum of a spatial representativeness error variance ($\sigma_s^2$) and a retrieval error variance ($\sigma_r^2$). The $\sigma_r$ value is the mean error estimated for the MUSICA IASI data within a IsoGSM grid box (it is typically 0.12 g/kg, 1.0 K, and 10‰ for $q$, $T$, and $\delta$D, respectively, Diekmann et al., 2021a). For the $\sigma_s$ values we use the standard deviations of the MUSICA IASI data within the IsoGSM grid box, which is generally of a similar order as $\sigma_r$. Table 1 gives a summary of the typically assumed observational errors.

In addition to the nature data, the data belonging to the different ensemble members have to be simulated. This is done with IsoGSM but with initialisations from different time steps within the same season of the nature run. These initial conditions are considered independent from the nature run (for more details see Toride et al., 2021). We calculate an ensemble with 96 members (i.e. $N_{\mathrm{ens}} = 96$).

**Table 1.** Table summarizing the typical free tropospheric observational error ($\sigma_o$) used for the assimilation experiments. The $\sigma_o$ values are calculated as the root-squares-sum of MUSICA IASI retrieval noise error ($\sigma_r$) and the spatial representativeness error ($\sigma_s$, due to resampling the small MUSICA IASI ground pixels onto the relatively coarse $200 \times 200$ km IsoGSM grid).

| Observation | $\sigma_o$ | $\sigma_r$ | $\sigma_s$ |
|---|---|---|---|
| $q$ | 0.30 g/kg | 0.12 g/kg | 0.27 g/kg |
| $T$ | 1.2 K | 1.0 K | 0.7 K |
| $\delta$D | 14‰ | 10‰ | 10‰ |

## 2.2 Data assimilation with a Kalman filter

For the data assimilation we use the Local Ensemble Transform Kalman Filter (LETKF, e.g. Hunt et al., 2007) method as developed for its use with water isotopologue data by Yoshimura et al. (2014). The Kalman filter based data assimilation technique optimally combines a model forecast with an observation by considering the respective model and observational uncertainties (Kalman, 1960). The result is a best estimate of the atmospheric state (the analysed state vector, $\boldsymbol{x}^a$):

$$\boldsymbol{x}^a = \boldsymbol{x}^b + \mathbf{K}(\boldsymbol{y} - \mathbf{H}\boldsymbol{x}^b), \tag{2}$$

where $\boldsymbol{x}^b$ is the so-called background state (the model forecast), $\boldsymbol{y}$ the observation vector, and $\mathbf{H}$ the observational operator (a matrix operator which maps the model state into the observation space). The matrix operator $\mathbf{K}$ is the Kalman gain:

$$\begin{aligned} \mathbf{K} &= \mathbf{B}\mathbf{H}^T(\mathbf{H}\mathbf{B}\mathbf{H}^T + \mathbf{R})^{-1} \\ &= (\mathbf{H}^T\mathbf{R}^{-1}\mathbf{H} + \mathbf{B}^{-1})^{-1}\mathbf{H}^T\mathbf{R}^{-1}, \end{aligned} \tag{3}$$

where the first and second line are the so-called $m$- and $n$-forms, respectively (whose equivalence is shown, for instance, in Chapt. 4 of Rodgers, 2000). The matrix $\mathbf{B}$ is the uncertainty covariance of the background state (it is calculated as the covariance of the different ensemble runs and thus captures the uncertainty of the model forecasts). Its inverse ($\mathbf{B}^{-1}$) is in the following also referred to as the background knowledge information matrix (it is a measure for the knowledge about the atmospheric state including the statistical dependency of different atmospheric state variables). The matrix $\mathbf{R}$ is the uncertainty covariance of the observational state (it captures the uncertainties of the observations). If we substitute in Eq. (2) $\mathbf{K}$ by the second line of Eq. (3) and $\boldsymbol{y}$ by $\mathbf{H}\boldsymbol{x}$ (the observation $\boldsymbol{y}$ is the actual atmospheric state $\boldsymbol{x}$ mapped to the observational domain) we get:

$$\boldsymbol{x}^a = \boldsymbol{x}^b + (\mathbf{H}^T\mathbf{R}^{-1}\mathbf{H} + \mathbf{B}^{-1})^{-1}\mathbf{H}^T\mathbf{R}^{-1}\mathbf{H}(\boldsymbol{x} - \boldsymbol{x}^b), \tag{4}$$

which reveals that the Kalman filter weights the impact of the background and the observation on the analyses reciprocally according to their respective uncertainties. More details on the used LETKF settings, like the localization, the covariance inflation or ensemble size choice, are given in Text S2 of the supplement of Toride et al. (2021).

**Table 2.** Table summarizing the different assimilation experiments used in this study. The column "Assimilated observations" lists the observations used for the experiment, the column "Symbol" shows the symbol used in the following when referring the respective experiment, the column "Corresponding $\Delta_{i,j}$" shows the symbol used for the corresponding $\Delta_{i,j}$ value, calculated according to Eq. (6), and the column "Corresponding RMSD" shows the symbol used for the corresponding RMSD value, calculated according to Eqs. (7).

| Assimilated observations | Symbol | Corresponding $\Delta_{i,j}$ | Corresponding RMSD |
|---|---|---|---|
| No observations | $\{\}$ | $\Delta_{i,j}\{\}$ | $\mathrm{RMSD}\{\}$ |
| $q$ | $\{q\}$ | $\Delta_{i,j}\{q\}$ | $\mathrm{RMSD}\{q\}$ |
| $q$ and $T$ | $\{q,T\}$ | $\Delta_{i,j}\{q,T\}$ | $\mathrm{RMSD}\{q,T\}$ |
| $q$ and $\delta$D | $\{q,\delta\mathrm{D}\}$ | $\Delta_{i,j}\{q,\delta\mathrm{D}\}$ | $\mathrm{RMSD}\{q,\delta\mathrm{D}\}$ |
| $T$ and $\delta$D | $\{T,\delta\mathrm{D}\}$ | $\Delta_{i,j}\{T,\delta\mathrm{D}\}$ | $\mathrm{RMSD}\{T,\delta\mathrm{D}\}$ |
| $q$, T and $\delta$D | $\{q,T,\delta\mathrm{D}\}$ | $\Delta_{i,j}\{q,T,\delta\mathrm{D}\}$ | $\mathrm{RMSD}\{q,T,\delta\mathrm{D}\}$ |

Our assimilation experiments use observations of specific humidity ($q$), atmospheric temperature ($T$) and the isotopologue ratio of water vapour ($\delta$D) at about $550\,\mathrm{hPa}$, which is the pressure level, where the MUSICA IASI products have generally a very good quality (high sensitivity and low uncertainty). An overview on the performed different assimilation experiments is given by Table 2.

### 2.3 Evaluation of the analyses quality

In the assimilation step the ensemble members are corrected according to the information provided by the observation. This results in an ensemble of analysed data. For convenience we interpolate the analyses fields to a regular $2.5° \times 2.5°$ horizontal grid and to 17 vertical pressure levels between 1000 and $10\,\mathrm{hPa}$. We use the mean value of these analysed data (i.e. the ensemble mean values) as representative for the analysis. For a time step $i$ and a location $j$ this ensemble mean is

$$\bar{x}_{i,j} = \frac{1}{N_{\mathrm{ens}}} \sum_{m=1}^{N_{\mathrm{ens}}} x_{m_{i,j}}, \tag{5}$$

where $x_{m_{i,j}}$ is the ensemble member $m$ at time step $i$ and location $j$. For each location and time step (each event) we calculate the difference of the ensemble mean and the nature data.

$$\Delta_{i,j} \quad = \quad \bar{x}_{i,j} - x_{n_{i,j}}. \tag{6}$$

This $\Delta_{i,j}$ captures the error for the single event corresponding to time step $i$ and location $j$. This is what we want to evaluate.

We then calculate the root-mean-squares of the $\Delta_{i,j}$ errors (root-mean-squares-differences, RMSD) for all events belonging to a group of events $A$:

$$\mathrm{RMSD} \quad = \quad \sqrt{\sum_{(i,j)\in A} \Delta_{i,j}^2 \Big/ \sum_{(i,j)\in A} 1}. \tag{7}$$

**Table 3.** Table with skill values discussed in this study. The column "Description of skill" outlines which assimilation experiments are used for calculating the skills (the evaluated experiment and the reference experiment, with respect to which the evaluation is performed), the column "Symbol" shows the symbol used in the text when referring to the respective skill, and the column "Definition of skill" describes how the skill is calculated according to Eq. (8).

| Description of skill | Symbol | Definition of skill |
|---|---|---|
| $\{q\}$ wrt $\{\}$ | $\{q\}_{\{\}}$ | $\frac{\mathrm{RMSD}\{\}-\mathrm{RMSD}\{q\}}{\mathrm{RMSD}\{\}}$ |
| $\{q,T\}$ wrt $\{\}$ | $\{q,T\}_{\{\}}$ | $\frac{\mathrm{RMSD}\{\}-\mathrm{RMSD}\{q,T\}}{\mathrm{RMSD}\{\}}$ |
| $\{q,T,\delta\mathrm{D}\}$ wrt $\{\}$ | $\{q,T,\delta\mathrm{D}\}_{\{\}}$ | $\frac{\mathrm{RMSD}\{\}-\mathrm{RMSD}\{q,T,\delta\mathrm{D}\}}{\mathrm{RMSD}\{\}}$ |
| $\{T,\delta\mathrm{D}\}$ wrt $\{q,T,\delta\mathrm{D}\}$, i.e. the $q$ observation impact | $\{T,\delta\mathrm{D}\}_{\{q,T,\delta\mathrm{D}\}}$ | $\frac{\mathrm{RMSD}\{q,T,\delta\mathrm{D}\}-\mathrm{RMSD}\{T,\delta\mathrm{D}\}}{\mathrm{RMSD}\{q,T,\delta\mathrm{D}\}}$ |
| $\{q,\delta\mathrm{D}\}$ wrt $\{q,T,\delta\mathrm{D}\}$, i.e. the $T$ observation impact | $\{q,\delta\mathrm{D}\}_{\{q,T,\delta\mathrm{D}\}}$ | $\frac{\mathrm{RMSD}\{q,T,\delta\mathrm{D}\}-\mathrm{RMSD}\{q,\delta\mathrm{D}\}}{\mathrm{RMSD}\{q,T,\delta\mathrm{D}\}}$ |
| $\{q,T\}$ wrt $\{q,T,\delta\mathrm{D}\}$, i.e. the $\delta\mathrm{D}$ observation impact | $\{q,T\}_{\{q,T,\delta\mathrm{D}\}}$ | $\frac{\mathrm{RMSD}\{q,T,\delta\mathrm{D}\}-\mathrm{RMSD}\{q,T\}}{\mathrm{RMSD}\{q,T,\delta\mathrm{D}\}}$ |

The group of events $A$ can include all events (sum over all time steps and locations) or only selected events that fulfil certain criteria. The RMSD values are a statistically robust metric representing the uncertainty of the analyses data for the events that belong to the group of events $A$.

From the RMSD values we then determine the skill of an assimilation experiment as:

$$\mathrm{Skill} = \frac{\mathrm{RMSD}\{\mathrm{ref}\} - \mathrm{RMSD}\{\mathrm{exp}\}}{\mathrm{RMSD}\{\mathrm{ref}\}}, \tag{8}$$

where $\mathrm{RMSD}\{\mathrm{exp}\}$ corresponds to the experiment we want to evaluate and $\mathrm{RMSD}\{\mathrm{ref}\}$ to the reference experiment with respect to which we want to do the evaluation. So the skill informs about the relative reduction of the RMSD value obtained from an assimilation experiment with respect to a reference assimilation experiment. We use the skill value throughout the paper for evaluating the quality of the different assimilation experiments.

We calculate two different types of skill values. For the first type, we use no data assimilation as the reference assimilation experiment. Here positive values document the relative improvement of the analyses when assimilating observations with respect to using no observations (a reduction of the RMSD value by the assimilation of the observations). The first three items in Table 3 correspond to the respective skill values used in this study. For the second type, we use the assimilation of all observations ($q$, $T$, and $\delta\mathrm{D}$ together) as the reference, and estimate the degradation of the analyses quality (an increase of the RMSD value or a "loss of skill") by removing one type of observation. By doing so, we can quantitatively compare the impact of the different observation types on the analyses quality. In the following we refer to this skill value as the observation impact, where a more negative value corresponds to a stronger observation impact. The respective skill values that are discussed in this study are listed as the three last items in Table 3.

The $Q_2$ values are calculated according to the budget analysis (Yanai et al., 1973):

$$Q_2 = -L\left(\frac{\partial q}{\partial t} + \boldsymbol{v}\cdot\nabla q + \omega\frac{\partial q}{\partial p}\right), \tag{9}$$

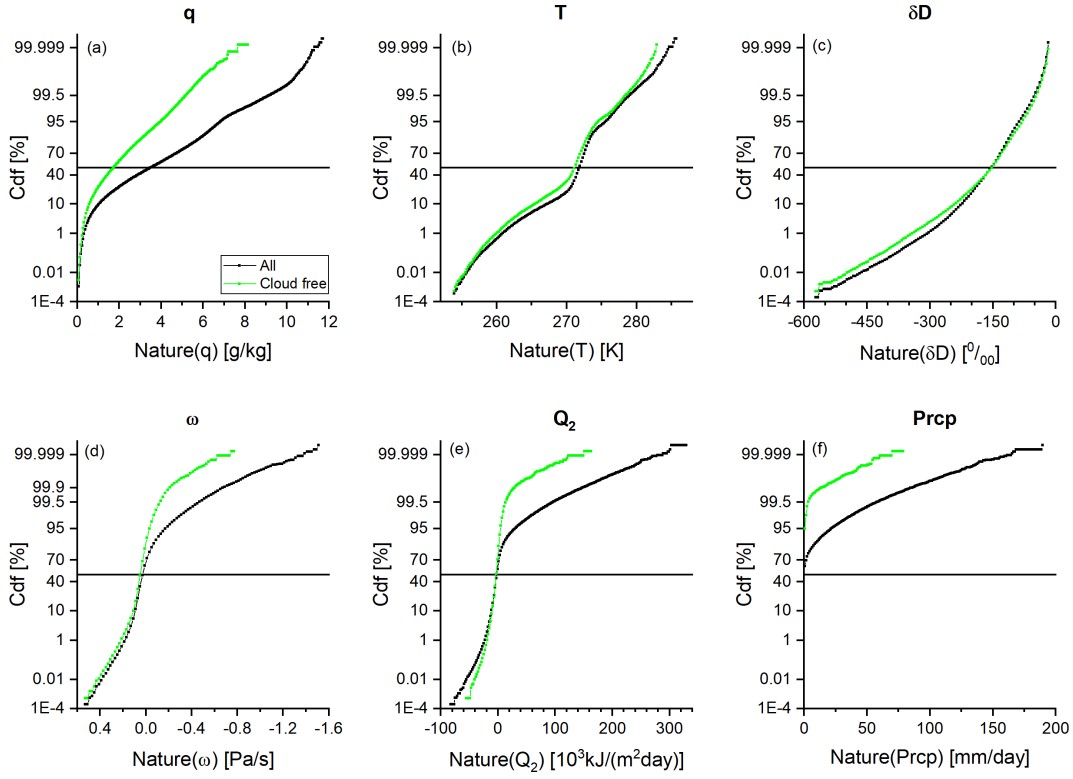

**Figure 1.** Cumulative distribution functions (cdfs) of the analysed parameters as obtained from all nature data (black) and from the nature data belonging to cloud-free events (green). (a) For specific humidity ($q$), (b) for temperature ($T$), (c) for the isotopologue ratio ($\delta$D), (d) for vertical velocity ($\omega$), (e) for latent heating ($Q_2$), and (f) for precipitation (Prcp). The 50th percentile is indicated by the black line.

where $L$ is the latent heat of net condensation, $q$ is the specific humidity, $\boldsymbol{v}$ is the horizontal wind vector, $\omega$ is the vertical velocity, and $p$ is the pressure. Please note that the so-calculated $Q_2$ might be slightly different from the real latent heating rates, because it misinterprets changes in specific humidity caused by sub-scale transport (like small-scale turbulent mixing or diffusion) as latent heat release/consumption.

We work with 6 hourly analyses data (for the 30°S - 30°N region) of July and August 2016. The ensemble simulations are made using 96 different initial conditions. The ensemble mean at the beginning of the simulation period represents climatology. The first three weeks of the simulation (beginning of July) is a "spin-up" period, when the analyses gradually approximate the nature data by assimilating enough observations. In order to avoid impacts of this "spin-up" period, respective data are excluded from the the evaluation study, which is then made for the mid-July to end of August period (covering 40.75 days).

Figure 1 shows the cumulative distribution functions (cdfs) of the variables $q$, $T$, $\delta$D, $\omega$, $Q_2$, and Prcp, calculated for the evaluated period from the nature data for all time steps and locations (black, full data set) and from the nature data belonging to a location and time step for which we can assimilate observations (green, data subset representing cloud-free conditions only). Only for $T$ and $\delta$D, the full data set and the cloud-free data subset show similar cdfs. Concerning $q$, $\omega$, $Q_2$, and Prcp, the

respective cdfs are significantly different. For $q$, all percentiles in the cloud-free data subset are shifted towards drier values if compared to the full data set. For $\omega$, $Q_2$, and Prcp, all distributions up to 50th percentiles are comparable for the full data set and the data subset; however, for large percentiles the two cdfs differ significantly. The most extreme values (very low $\omega$, very high $Q_2$ and Prcp) are not present in the cloud-free data subset. This shows that we do not assimilate observations that directly represent the atmosphere of the locations and time steps where these extreme events take place.

## 3   Overview on assimilation impacts

This section gives an overview on the assimilation impacts. For this purpose we calculate the RMSD values using all events (averaging is performed over all time steps and locations). Equation (7) can then be written as:

$$\text{RMSD} \quad = \quad \sqrt{\frac{1}{N_{\text{loc}}N_{\text{tim}}}\sum_{j=1}^{N_{\text{loc}}}\sum_{i=1}^{N_{\text{tim}}}\Delta_{i,j}^2}, \tag{10}$$

where $N_{\text{loc}}$ is the number of all locations (here we investigate the 30°S-30°N region with a $2.5° \times 2.5°$ resolution, i.e. $N_{\text{loc}} = 3600$), and $N_{\text{tim}}$ is the number of all time steps (here we work with 6 h time steps covering 40.75 days, i.e. $N_{\text{tim}} = 163$).

Because continuous time series are used for this calculation, we cannot assume independence of the different data when estimating the uncertainty of the RMSD values. For this reason, we use the circular block bootstrap method (e.g. Wilks, 2019) for the RMSD uncertainty estimation (the method is also explained in the supplement of Toride et al., 2021). We resample these data 10000 times, which provides a representative distribution of possible RMSD values. Here we use the half of the difference between the respective 15.9th and 84.1th percentile estimates as the $1\sigma$ uncertainty of the RMSD value, which we then propagate to the skill values.

### 3.1   Skills with respect to no data assimilation

Observations of free tropospheric $q$ and $T$ contain important information on the atmospheric state (among others on the water cycle variables $\omega$, $Q_2$, and Prcp) and are available as standard products from different satellite data processors at global scale, daily coverage, and with good precision. Recently, respective observations of free tropospheric $\delta$D have become available. We use the experiments that assimilate these observations in order to understand to what extent these observations help to constrain the model uncertainty.

First we calculate the skills achieved when assimilating $q$ observations only using no data assimilation as the reference, i.e. here $\text{RMSD}_{\text{ref}}$ of Eq. (8) is for ensemble means ($\bar{x}_{i,j}$) obtained without assimilating any observation (no data assimilation step). This skill is referred to in the following as the $\{q\}_{\{\}}$ skill (see Table 3). The black lines in Figure 2a-e show the vertical dependency of the $\{q\}_{\{\}}$ skills (for Prcp there is naturally no vertical dependency, Fig. 2f). The grey area around the lines indicates the $1\sigma$ uncertainty of the skills. Because we assimilate the observations of $q$ at about 550 hPa, highest skill values are generally achieved in the free troposphere around 550 hPa. The dotted lines correspond to the pressure levels at 775 and 350 hPa, which delimits the vertical range we use as representative for the free troposphere and for which we perform dedicated evaluations in Sect. 4.

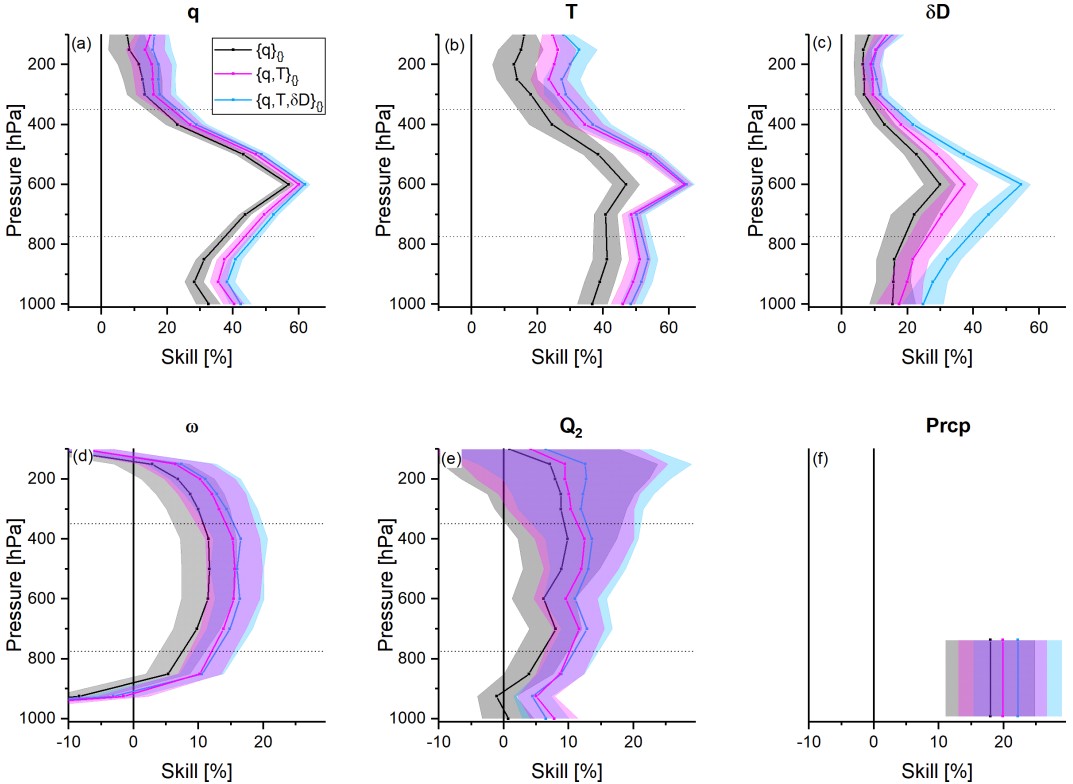

**Figure 2.** Vertical profiles of the skills achieved by assimilating the observations (only $q$, $q$ and $T$, and $q$, $T$, and $\delta$D) versus assimilating no observations. Black/grey: $\{q\}_{\{\}}$ skill; magenta: $\{q,T\}_{\{\}}$ skill; light blue: $\{q,T,\delta$D$\}_{\{\}}$ skill. The area around the lines represents the $1\sigma$ uncertainty. (a) For specific humidity ($q$), (b) for temperature ($T$), (c) for the isotopologue ratio ($\delta$D), (d) for vertical velocity ($\omega$), (e) for latent heating ($Q_2$), and (f) for precipitation (Prcp).

In a second experiment we assimilate observations of $q$ together with $T$, which comes very close to an assimilation of relative humidity data. For the evaluation we again calculate the skills with respect to no data assimilation (in the following referred to as the $\{q,T\}_{\{\}}$ skill, see Table 3). The magenta lines in Figs. 2a-f give an overview on the achieved $\{q,T\}_{\{\}}$ skills. Compared to the $\{q\}_{\{\}}$ skills, these skills are larger in particular for $T$ around 550 hPa (Fig. 2b), because $T$ at 550 hPa is the additional observation we assimilate. The additional assimilation of $T$ has also positive impacts on $q$ and $Q_2$ above 700 hPa and on $\delta$D

between 500 and 300 hPa. By assimilating the standard observations $q$ and $T$, we achieve skills of up to 60% for $q$ and $T$ around 600 hPa. Also for the other variables ($\delta$D, $\omega$, $Q_2$, and Prcp) – for which no respective observations are assimilated – we get skills that are often between 10% and 30%.

In a third experiment we test the impact when assimilating $q$, $T$, and $\delta$D observations together. For the evaluation we again calculate the skills with respect to no data assimilation (in the following referred to as the $\{q,T,\delta$D$\}_{\{\}}$ skill, respectively,

see Table 3). The bright blue lines in Figs. 2a-f give an overview on the achieved $\{q,T,\delta$D$\}_{\{\}}$ skills. The values are further improved if compared to the $\{q,T\}_{\{\}}$ skills: significantly for the $\delta$D (Fig. 2c, the respective $\delta$D skill is above 50% at 600 hPa),

but only slightly for the other variables. By significant, we mean that the skill values are larger than the estimated $1\sigma$ uncertainty of the skills, which is represented by the shaded area around the lines.

## 3.2 Observation impacts of $q$, $T$, and $\delta$D

Figure 2 reveals that assimilating $q$, $T$ and $\delta$D observations at about $550\,$hPa well constrains the uncertainty of free tropospheric $q$, $T$, and $\delta$D simulations between about 775 and $350\,$hPa (skill values up to 60%). There is also a significant improvement for the simulated water cycle variables $\omega$, $Q_2$, and Prcp (skill values above 20%). In this subsection, we examine the importance of the different types of observation ($q$, $T$, and $\delta$D, respectively) for achieving this analyses quality.

   For this purpose we use the experiment that assimilates $q$, $T$, and $\delta$D observations together as the reference, i.e. $\mathrm{RMSD}_{\mathrm{ref}}$
of Eq. (8) is for ensemble means ($\bar{x}_{i,j}$) obtained when assimilating observations of $q$, $T$, and $\delta$D together. Then we compare this reference to an experiment for which one observation type has been removed and calculate the respective "loss of skill" values (see last three items of Table 3). A large negative "loss of skill" value means that the respective observation is very important for achieving the analyses quality of the reference experiment, i.e. the respective observation has a strong impact on the analyses quality. Our particular interest is in comparing the impact of $\delta$D observations, that have become only recently
available on global scale and daily coverage, to the respective impacts of the traditionally used observations of $q$ and $T$.

   Figure 3a-f shows these observation impacts. We first evaluate the impact of the $q$ observations, which we determine by calculating the loss of skill when assimilating only observations of $T$ and $\delta$D instead of assimilating all three observation types (i.e. the $q$ observation impact is quantified by the $\{T,\delta\mathrm{D}\}_{\{q,T,\delta\mathrm{D}\}}$ skill value, see Table 3). The $q$ observation impacts are represented by the dark yellow line (and the shaded area is the respective $1\sigma$ uncertainty). We observe that, when removing $q$
observations, we lose a lot of skill for all atmospheric variables, i.e. the $q$ observations are important and have a strong impact on all the analysed variables. For the analyses of $q$ the impact is strongest at 500-600 hPa (loss of skill of up to $-50\%$); for other vertical pressure levels the impact is smaller with loss of skill values above or close to $-20\%$ (Fig. 3a). There is also a significant impact of $q$ observations on the analyses of $T$, $\delta$D, $Q_2$, $\omega$ and Prcp (although less than for $q$): the $\{T,\delta\mathrm{D}\}_{\{q,T,\delta\mathrm{D}\}}$ skills are close to $-20\%$ above 700 hPa for $T$ (Fig. 3b), close to $-15\%$ around 600 hPa for $\delta$D (Figs. 3c) and close to $-10\%$
for free tropospheric $\omega$ and $Q_2$ (Figs. 3d+e) as well as for Prcp (Figs. 3f). By significant, we mean that the calculated loss of skill value is smaller than the estimated $1\sigma$ uncertainty of the skill, which is represented by the shaded area around the red line.

   The red line in Fig. 3 represents the $T$ observation impact, which we quantify by the "loss of skill" when assimilating only observations of $q$ and $\delta$D instead of assimilating all three observation types (the $\{q,\delta\mathrm{D}\}_{\{q,T,\delta\mathrm{D}\}}$ skill value, see Table 3). The
$T$ observation impact is highest for $T$. At 500-600 hPa the respective loss of skill values are about $-40\%$ and for other vertical pressure levels they are close to $-15\%$ (Fig. 3b). The $T$ observations have also a significant impact on the analyses of $q$, $\delta$D, $Q_2$, $\omega$ and Prcp (although less than for $T$): the $\{q,\delta\mathrm{D}\}_{\{q,T,\delta\mathrm{D}\}}$ skill values are close to $-10\%$ above 600 hPa for $q$ (Fig. 3a) and close to $-10\%$ at 500-700 hPa for $\delta$D (Fig. 3c). For free tropospheric $\omega$ and $Q_2$ as well as for Prcp the values are still about $-5\%$ (Figs. 3d-f). However, for Prcp this impact is not significant.

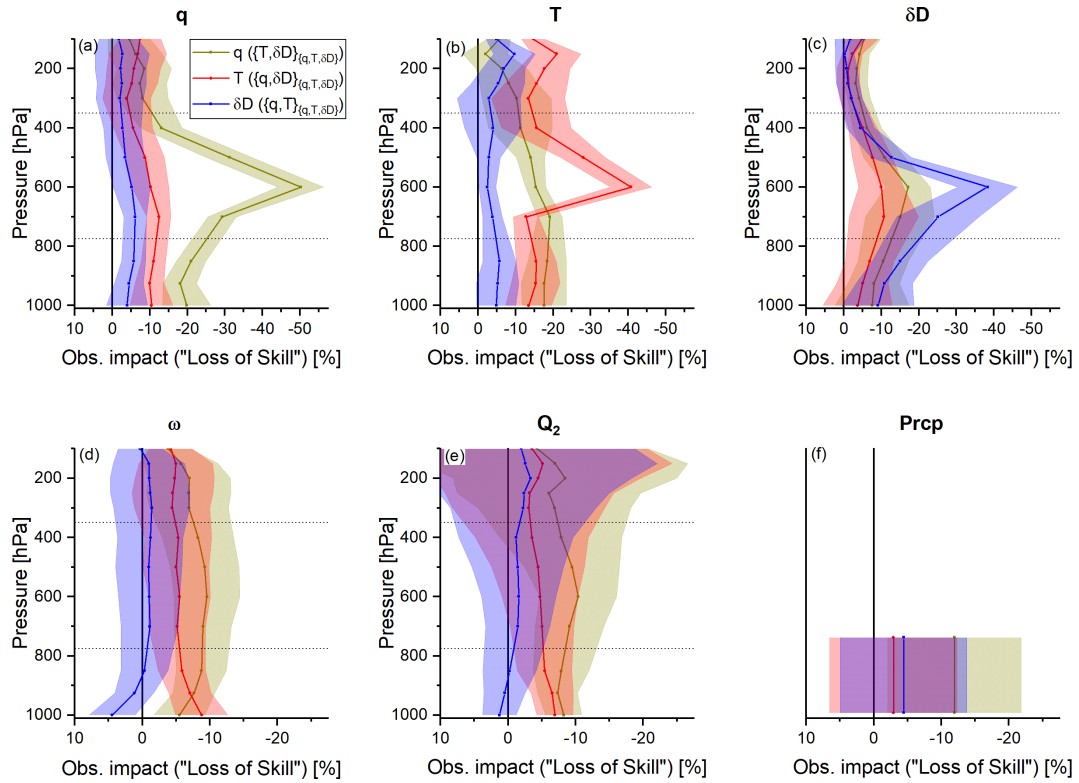

**Figure 3.** Vertical profiles of observation impacts ("loss of skill" by removing one observation type from the reference experiment that considers all three observation types). Dark yellow: impact of $q$, represented by the $\{T, \delta D\}_{\{q,T,\delta D\}}$ skill; red: impact of T, represented by the $\{q, \delta D\}_{\{q,T,\delta D\}}$ skill; blue: impact of $\delta D$, represented by the $\{q, T\}_{\{q,T,\delta D\}}$ skill. The panels (a)-(f) represent the different atmospheric variables as in Fig. 2.

In a final setup we investigate the loss of skills when assimilating only observations of $q$ and $T$ instead of all three observation types, i.e. we calculate the $\{q, T\}_{\{q,T,\delta D\}}$ skill as a measure for the $\delta D$ observation impact (see Table 3). The overview for the $\delta D$ observation impact is shown as blue lines in Fig. 3. The strongest impact is observed for the $\delta D$ analyses with respective loss of skill values of close to $-40\%$ around $600\,\text{hPa}$, and about $-10$ to $-30\%$ for other pressure levels above $400\,\text{hPa}$ (Fig. 3c), which is reasonable because observation of $\delta D$ are important for achieving a high quality of the $\delta D$ analyses. Concerning the $q$

and $T$ analyses, the $\delta D$ observations have a significant impact above $500\,\text{hPa}$ (for $900$-$500\,\text{hPa}$ the loss of skill values are about $-5\%$). For lower pressure levels the $\delta D$ observation impact on the $q$ and $T$ analyses is not significant (the loss of skill value is smaller than the estimated uncertainty, Fig. 3a+b). For $\omega$ and $Q_2$, the $\delta D$ observation impact is small and not significant ($\{q, T\}_{\{q,T,\delta D\}}$ skills of about $-2.5\%$ only, Figs. 3d+e). For Prcp, we observe a loss of skill value of $-5\%$, which suggests that the $\delta D$ observation impact on the Prcp analyses is slightly stronger than the respective $T$ observation impact; however, it

is not significant (compare red and blue lines in Fig. 3f).

## 4 The complementarity of $\delta$D observations

The overview study of the previous section reveals that the $\delta$D observation impact is overall weak and generally much smaller than the respective impacts of the $q$ and $T$ observations. Theoretically the $\delta$D data contain unique information on phase transitions, i.e. it might be expected that $\delta$D observations can in particular improve the quality of the analyses for atmospheric conditions that involve strong and/or repeated cycles of condensation (or evaporation) processes. In this section we examine the adequacy of this hypothesis in more detail and focus on the analyses of data averaged over a free tropospheric pressure range (775-350 hPa, indicated by the dotted lines in Figs. 2 and 3).

### 4.1 Analyses quality and vertical velocity

For vertically unstable atmospheric conditions repeated cycles of condensation and evaporation take place. For this reason we can examine the aforementioned hypothesis by investigating the dependency of the $\delta$D observation impact on atmospheric vertical stability and we use the mass weighted average between 775 and 350 hPa of vertical velocity (free tropospheric $\omega$) as a proxy for atmospheric vertical stability.

Figure 4 depicts the dependency of the free tropospheric analyses errors (the $\Delta_{i,j}$ values, see Eq. (6)) on the free tropospheric $\omega$ as simulated by the nature run ($\omega_{\mathrm{nat}}$). As in Figs. 2 and 3, we investigate the analyses of the atmospheric variables $q$, $T$, $\delta$D, $\omega$, $Q_2$, and Prcp. We examine the low latitudes (30°S - 30°N, with a $2.5° \times 2.5°$ horizontal resolution) for 40 days with a 6 hourly time resolution, i.e. in total we have 586800 events. In order to visualize the distribution of this large amount of data points, we calculate the data densities as follows: we generate 60 equidistant $\omega$-bins covering all occurring $\omega$. Then we calculate the density distribution of the $\Delta_{i,j}$ values in each $\omega$-bin and sum up the number of data points belonging to the highest $\Delta_{i,j}$ densities until we consider 98% of all the $\Delta_{i,j}$ values occurring for a particular $\omega$-bin. These 98% areas are depicted in Fig. 4. The grey filled area represents the $\Delta\{\}$ distribution (i.e. for the $\Delta_{i,j}$ values when no observations are assimilated), and the magenta and blue lines the 98% contour lines for the $\Delta\{q,T\}$ and $\Delta\{q,T,\delta\mathrm{D}\}$ distributions (i.e. for the $\Delta_{i,j}$ values achieved when we assimilate $q$ and $T$ observations and $q$, $T$ and $\delta$D observations, respectively).

Figures 4a-c show the distributions of the $\Delta_{i,j}$ values for the variables $q$, $T$, and $\delta$D. There is a weak correlation between the analyses errors and the $\omega_{\mathrm{nat}}$ data, i.e. the $\Delta_{i,j}$ values for $q$ tend to be positive for stable atmospheric conditions (positive $\omega$) and negative for unstable atmospheric conditions (strongly negative $\omega$). A weak dependency is also observed in the $\Delta_{i,j}$ for $T$ and to an even smaller extent in the $\Delta_{i,j}$ values for $\delta$D: in both cases the largest positive values occur for stable conditions (positive $\omega$) and negative values are more frequent for unstable conditions (negative $\omega$). The assimilation of $q$ and $T$ observations approximates the $\Delta_{i,j}$ values to the respective $\Delta$-zero lines. Concerning the $q$ analyses the additional assimilation of $\delta$D observations further reduces the error in particular for strongly unstable conditions (for highly negative $\omega_{\mathrm{nat}}$ the blue contour line better approximates the $\Delta$-zero line than the magenta contour line, Fig. 4a). For $T$ the additional impact when assimilating $\delta$D observation is also slightly larger for unstable conditions (Fig. 4b). For $\delta$D the additional impact of assimilating $\delta$D observations is most pronounced for stable conditions (blue contour lines better approximates the $\Delta$-zero line than the magenta contour for positive $\omega$ values, Fig. 4c).

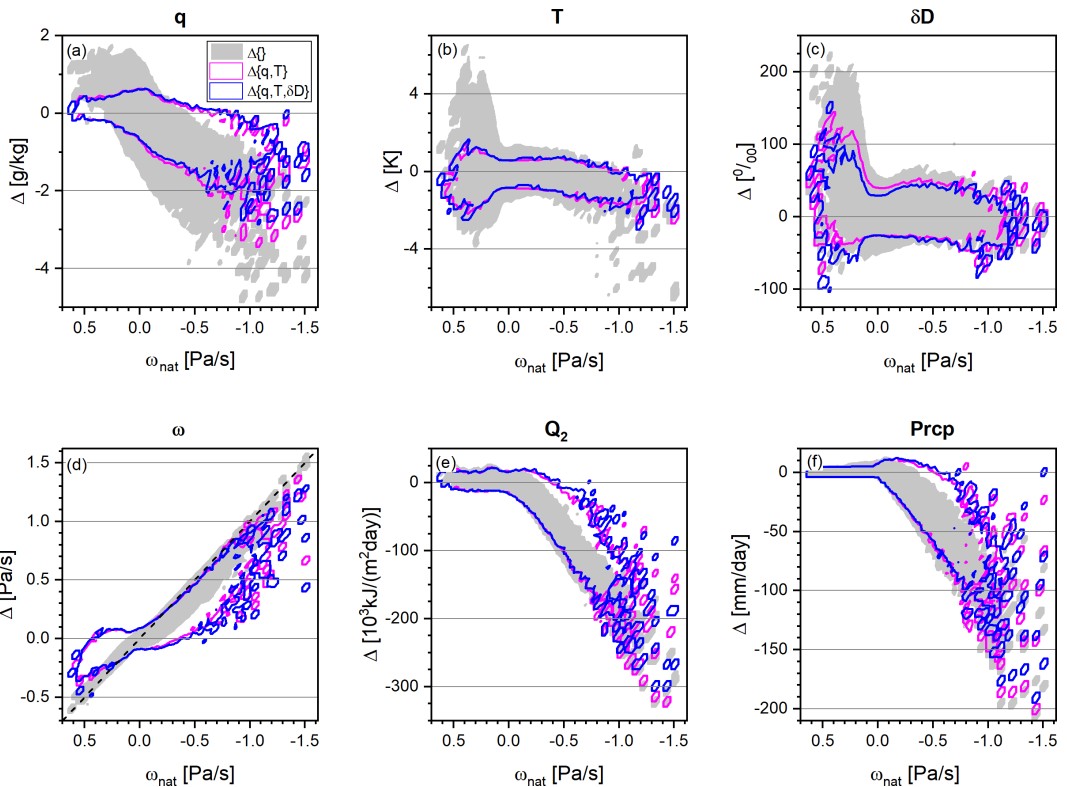

**Figure 4.** Dependency of the free tropospheric analyses errors (mass weighted averages between 775 and 350 hPa) on the free tropospheric vertical velocity ($\omega_{nat}$). Shown are the areas that contain 98% of all the $\Delta_{i,j}$ values for a given $\omega_{nat}$ value. Grey: 98% area for no data assimilation (the $\Delta\{\}_{i,j}$ data distribution). Magenta line: 98% contour line for the $\Delta\{q,T\}_{i,j}$ data distribution. Blue line: 98% contour line for the $\Delta\{q,T,\delta D\}_{i,j}$ data distribution. (a) for specific humidity ($q$), (b) for temperature ($T$), (c) for the isotopologue ratio ($\delta D$), (d) for vertical velocity ($\omega$), (e) for latent heating ($Q_2$), and (f) for precipitation (Prcp).

When no observations are assimilated, the strength of atmospheric stability can hardly be identified and the $\omega$ error has the same magnitude as the actual $\omega$ value (the $\Delta\{\}$ distribution in Fig. 4d aligns very closely with the black dashed diagonal). The $\Delta\{q,T\}$ and $\Delta\{q,T,\delta D\}$ distributions dissipate from the diagonal and approximate better the $\Delta$-zero line. This reduction of the $\omega$ error is slightly more pronounced for the $\Delta\{q,T,\delta D\}$ than for the $\Delta\{q,T\}$ distributions, and largest for the most negative actual $\omega$ values. However, despite the significant correction, the error is still largest for the most negative $\omega$ values. This means that, although the events of vertically unstable atmospheric conditions can be much better identified by assimilating the observations, the absolute strength of the instability is still underestimated.

For the analyses of latent heating ($Q_2$, Fig. 4e) and precipitation rate (Prcp, Fig. 4f), the results are very similar to those of $\omega$: without assimilating any data ($\Delta\{\}$ densities), the analyses are very uncertain for high heating rates (distribution of $\Delta_{i,j}$ values is far away from the respective $\Delta$-zero lines). Actually vertical velocity, precipitation rate, and heating rate are strongly correlated, which means that the events with strong latent heating and/or with high precipitation rates are almost not identified.

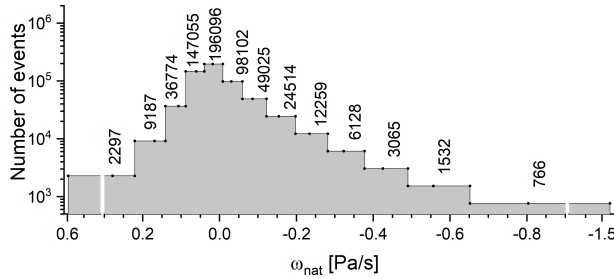

**Figure 5.** Abundance chart showing the number of events for each of the 13 $\omega$-bins used for classifying atmospheric vertical stability conditions.

By assimilating $q$ and $T$ observations, these errors can be strongly reduced. A further significant reduction (in particular for events when the error is very high) can be achieved by assimilating $\delta$D observations in addition to the observations of $q$ and $T$.

### 4.2 The unique $\delta$D assimilation impact

Figure 4 suggests that when assimilating $q$ and $T$ together with $\delta$D we get smaller analyses errors for unstable atmospheric conditions than when only assimilating $q$ and $T$. In this subsection we quantify how the observation impacts depend on the atmospheric vertical stability.

Figure 5 shows the abundances of events corresponding to 13 different free tropospheric vertical velocity ($\omega$) bins. We have the highest abundances for $\omega$ values that are close to zero. The three bins corresponding to $\omega_{\mathrm{nat}}$ values between $-0.09$ and $+0.06$ Pa/s comprise together 441253 out of 586800 events, which is 75.20%. Extreme vertical instabilities are rare, e.g. the bins corresponding to $\omega_{\mathrm{nat}}$ values smaller than $-0.38$ Pa/s only comprises 5363 events, i.e. 0.91%. However, these extreme events are responsible for almost all the intense precipitation events, and it is very important to improve the analyses in a way that allows a better identification of these extreme events.

We use the binning from Fig. 5 for evaluating the dependence of the observation impacts on $\omega$. As in Sect. 3, we quantify the observation impacts by the loss of skill values according to the last three items in Table 3. We calculate the respective RMSD values according to Eq. (7) for 13 different groups of events $A$. Each group comprises the events showing free tropospheric $\omega$ values as defined by the 13 different bins of Fig. 5. Because the events with strongly negative $\omega$ values (convective events) are generally individual events occurring on a single day, the respective groups of events do not consist of continuous time series. For the error estimation, we thus assume that the events of a certain $\omega$ group are independent (in the circular block bootstrap method the block size is reduced to only one event).

Figure 6 depicts the observation impacts obtained for the 13 $\omega_{\mathrm{nat}}$-bins. The colours are as in Fig. 3. The $q$ observation impact (quantified by the $\{T, \delta\mathrm{D}\}_{\{q,T,\delta\mathrm{D}\}}$ skill value, represented by the dark yellow lines) is strong for all analysed variables (as already shown in Fig. 3). The highest $q$ observation impact is found for $q$ analyses at stable atmospheric conditions (loss of skill value of about $-60\%$ for $\omega_{\mathrm{nat}} > +0.1$ Pa/s, Fig. 6a). For the $T$ and $\delta$D analyses the $q$ observation impact is strongest, and

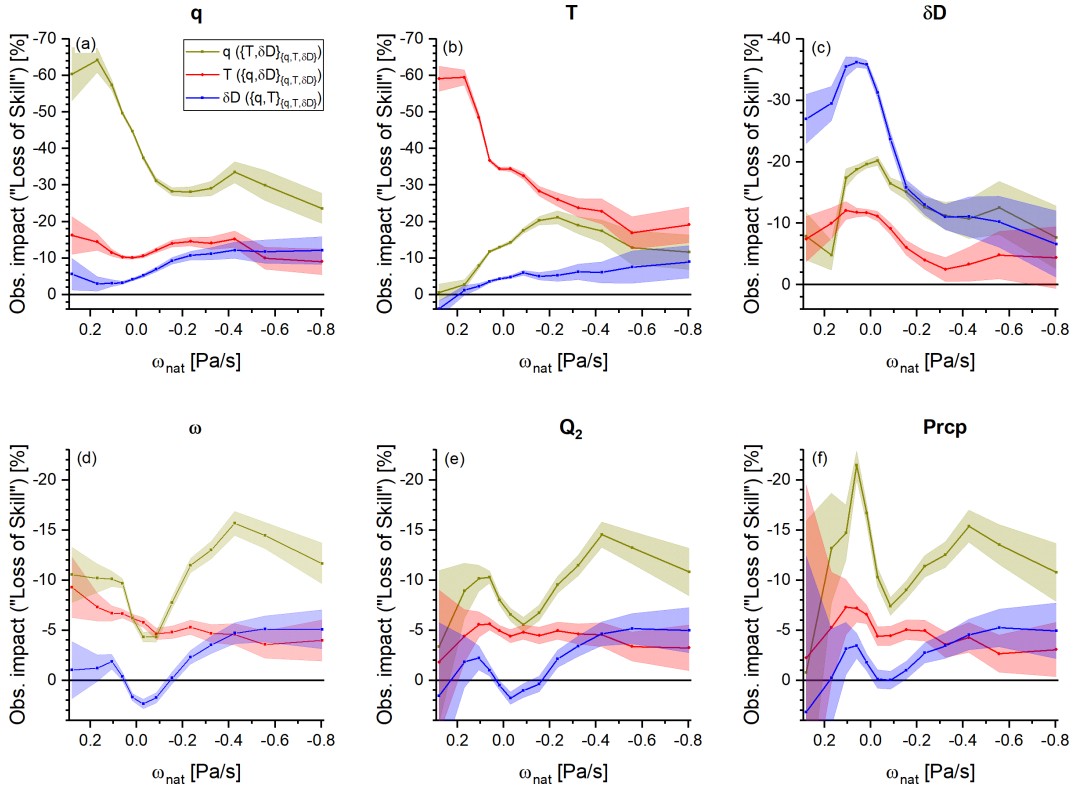

**Figure 6.** Dependency of the free tropospheric observation impacts (mass weighted averages between 775 and 350 hPa) on the free tropospheric vertical velocity ($\omega_{\mathrm{nat}}$). The colours (dark yellow, red, and blue) and the panels (a-f) are as in Fig. 3: they represent the different observation impacts and different analysed atmospheric variables, respectively.

for the analyses of $\omega$, $Q_2$, and Prcp, it is weakest for $\omega_{\mathrm{nat}}$ close to zero (Fig. 6b+c and Fig. 6d-f, respectively). In summary, there is no clear systematic dependency of the $q$ observation impact on atmospheric vertical stability.

The red lines represent the $T$ observation impact (quantified by the $\{q, \delta D\}_{\{q,T,\delta D\}}$ skill value). It is strongest for the $T$ analyses at stable atmospheric conditions (loss of skill value of about $-60\%$ for $\omega_{\mathrm{nat}} > +0.1$ Pa/s, Fig. 6b). Concerning the analyses of the other atmospheric variables, the $T$ observation impact shows generally a weak decrease with decreasing $\omega_{\mathrm{nat}}$, i.e. the impact tends to be slightly stronger for stable than unstable atmospheric conditions (Figs. 6a,c-f).

  The $\{q, T\}_{\{q,T,\delta D\}}$ skill values quantify the $\delta D$ observation impact and they are depicted as blue lines in Fig. 6. The strongest
$\delta D$ observation impact is found for the $\delta D$ analyses for stable atmospheric conditions (for $\omega_{\mathrm{nat}} > -0.05$ Pa/s the respective loss of skill value is beyond $-25\%$, Fig. 6c). For the analyses of all other variables, the $\delta D$ observation impact tends to be stronger for unstable if compared to stable atmospheric conditions (Fig. 6a,b,d-f), thus showing the opposite behaviour as the $T$ observation impact. While for moderately stable atmospheric conditions ($\omega_{\mathrm{nat}} > -0.2$ Pa/s), the $T$ observation impact is significantly stronger than the $\delta D$ observation impact, for unstable conditions ($\omega_{\mathrm{nat}} < -0.4$ Pa/s), the $\delta D$ observation impact
becomes as strong as the $T$ observation impact or even slightly exceeds it. Because the large majority of events correspond

to relatively stable atmospheric conditions ($\omega_{\mathrm{nat}} > -0.2\,\mathrm{Pa/s}$), the overview study as shown in Fig. 3 reveals an overall weak $\delta\mathrm{D}$ observation impact. However, for the infrequently occurring events corresponding to unstable atmospheric conditions, $\delta\mathrm{D}$ observations become at least as important as $T$ observations for all variables except for $T$ itself. This is particularly important for the analyses of $\omega$, $Q_2$, and Prcp, because the most extreme $\omega$, $Q_2$, and Prcp values are relatively poorly identified by assimilating only $q$ and $T$ observations; a better identification of these events is achieved by the additional assimilation of $\delta\mathrm{D}$ (compare the magenta and blue contour lines in Fig. 4d-f).

Moreover, the relatively strong $\delta\mathrm{D}$ observation impact occurs for conditions when there are no observations assimilated: a thermal infrared sensor like IASI offers no free tropospheric products for mid- or high-level clouds, which are typically present for $\omega_{\mathrm{nat}} < -0.4\,\mathrm{Pa/s}$ (see Fig. 1d). This suggests that the distinct free tropospheric $\delta\mathrm{D}$ signals caused by atmospheric convection (e.g. Risi et al., 2008; Diekmann et al., 2021b) are well conserved in the $\delta\mathrm{D}$ fields modelled for cloud-free locations outside of the convective area. At the cloud-free location the observations can then be exploited by the assimilation system and allow for improving the analyses of the convective area. The $\delta\mathrm{D}$ observations seem to have a unique remote impact on the analyses of convective regions.

### 4.3 Simulations versus real world data

In order to link our OSSE study to the real world, we examine similarities and differences of the $\{q, \delta\mathrm{D}\}$-pair distributions between the simulated data used in this study and actual MUSICA IASI observational data. The $\{q, \delta\mathrm{D}\}$-pair distributions can give valuable insight into the dominating atmospheric processes (mixing, shallow cloud formation and rain-out, convection and extreme precipitation events, Noone, 2012). Figure 7 shows these distributions for different data (sub-)sets. Shown are the areas where the $\{q, \delta\mathrm{D}\}$-pairs have the highest densities and sum up to 90% (thick contour lines) and 50% (thin contour lines) of all the data.

Concerning the OSSE data, the black lines show the distribution for the full nature run data set (586800 events for the studied 40 days and the 30°S-30°N area). The thick grey dotted line represents a typical tropical Rayleigh line (starting conditions: $T = 25°\mathrm{C}$, $\mathrm{RH} = 80\%$, and $\delta\mathrm{D} = -80‰$). A Rayleigh line describes the $\{q, \delta\mathrm{D}\}$ relation assuming that all the condensed water is immediately removed (immediate rain-out). We observe that the $\{q, \delta\mathrm{D}\}$-pairs are well distributed around the Rayleigh line. For dry conditions, the data points tend to lie above the Rayleigh line, which indicates that the respective humidity levels are largely controlled by mixing between humid/enriched and dry/depleted water masses (Noone et al., 2011; González et al., 2016). For humid conditions, the data points tend to be situated below the Rayleigh line, i.e. in the super-Rayleigh domain (Noone, 2012). This strong depletion together with high humidity is caused by recurring evaporation and condensation, i.e. the same water mass experiences several condensation/evaporation processes in the atmosphere. This is a typical free tropospheric $\{q, \delta\mathrm{D}\}$-pair signal for convective activity (e.g. Risi et al., 2008; Blossey et al., 2010; Noone, 2012; Diekmann et al., 2021b).

The green contour lines represent the distribution for the atmospheric conditions, for which observations are assimilated in our assimilation studies. These are the cloud-free and stable atmospheric conditions. For this data subset, the $\{q, \delta\mathrm{D}\}$-pairs are mostly located below $q = 6$ g/kg and above the Rayleigh line, caused by the aforementioned dry air mass mixing.

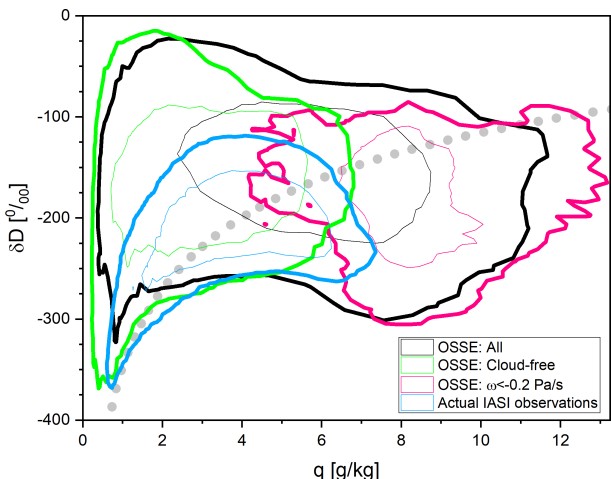

**Figure 7.** Distributions of $\{q, \delta D\}$-pairs at about 600 hPa as derived from the different data sets. Shown are the contour lines for the highest $\{q, \delta D\}$-pair data density (thick and thin lines show the areas containing 90% and 50% of all the data, respectively). Black line: all data from the nature run; green line: cloud-free data from the nature run, i.e. data used as observations during the assimilation step; pink line: nature run data corresponding to unstable atmospheric conditions ($\omega_{nat} < -0.2$ Pa/s); bright blue line: actual IASI observation data. The dotted grey line is a typical tropical Rayleigh line, assuming the following atmospheric condition over the ocean source location: $T = 25°C$, RH = 80%, and $\delta D = -80‰$.

The pink contour line comprises the events corresponding to unstable atmospheric conditions ($\omega_{nat} < -0.2$ Pa/s). The re-
spective $\{q, \delta D\}$-pairs are generally located in the aforementioned super-Rayleigh domain, which suggests convective activity. From Figs. 4 and 6 we can conclude that the $\delta D$ observations have the strongest impact on the analyses of the events with $\omega_{nat} < -0.2$ Pa/s, i.e. for events where the $\{q, \delta D\}$-pairs show this super-Rayleigh distribution. However, this is very different from the distribution of the assimilated data (green contour lines), revealing again that we do not assimilate observation of convective atmospheres (recall the discussion in the context of Fig. 1).

The $\{q, \delta D\}$-pair distribution obtained from actual MUSICA IASI observations is represented by the bright blue contour lines in Fig. 7 for the same period and locations as the OSSE data. This real world data is only available for a cloud-free atmosphere. Obviously, there is a significant difference between the $\{q, \delta D\}$-pair distribution simulated for cloud-free conditions and the actually observed distributions (compare green and bright blue contour lines in Fig. 7). Whereas the $q$ values in the simulations and the real world observations are very similar, the respective $\delta D$ values are systematically by about 50-100‰ lower
in the observations if compared to the simulations (which is significantly larger than the systematic uncertainty estimated for the MUSICA IASI data after its calibration to in-situ aircraft profiles, Schneider et al., 2016). While in the simulations the large majority of the $\{q, \delta D\}$-pairs is located above the Rayleigh line, in the observations about half is above and the other half is below the Rayleigh line, i.e. a super-Rayleigh distribution is regularly observed in the real world but very rarely in the simulations. Section 6.3 of Diekmann (2021) documents that the MUSICA IASI $\{q, \delta D\}$-pair data observed in the context of
the West African Monsoon is generally located below the Rayleigh line if a convective event was happening shortly before

the observation, which highlights the strong link between the regularly observed super-Rayleigh distributions and convective processes. This link seems to be weaker in the simulations, i.e. there the convective processes leave a significantly weaker $\{q, \delta D\}$-pair signature on the nearby cloud-free atmosphere.

## 4.4 Outlook on assimilating real world $\delta$D observations

Current state-of-the-art satellite sensors allow the observation of $\delta$D with high quality and resolution (e.g. Worden et al., 2007; Frankenberg et al., 2009; Schneider and Hase, 2011; Lacour et al., 2012; Boesch et al., 2013; Worden et al., 2019; Schneider et al., 2020; Diekmann et al., 2021a). Furthermore, $\{q, \delta D\}$-pair super-Rayleigh distributions have been observed in data sets generated from measurements of the TES (Tropospheric Emission Spectrometer) satellite instrument (e.g. Noone, 2012) and in the MUSICA IASI data (Schneider et al., 2017; Diekmann et al., 2021a; Diekmann, 2021). As discussed in the

previous subsection, these super-Rayleigh distributions contain valuable information on convective processes. In this context, the IASI instrument and thus the MUSICA IASI data set is in particular promising: measurements of IASI (or IASI-NG, the successor instrument of IASI) offer a very high horizontal and spatial coverage, and are guaranteed at least for the next two decades in the framework of the Metop and Metop-SG missions of EUMETSAT (European Organisation for the Exploitation of Meteorological Satellites, https://www.eumetsat.int/our-satellites/metop-series), i.e. respective $\{q, \delta D\}$-pair observations

are guaranteed for the next decades.

However, in order to optimally use these $\delta$D observations in data assimilation approaches, we need atmospheric isotopologue enabled models that capture as many details of a convective atmosphere as possible. The IsoGSM model used here with $200 \times 200$ km horizontal resolution does generally a good job, which has been demonstrated in different model validation studies (e.g. Yoshimura et al., 2008; Schneider et al., 2010). However, Fig. 7 reveals that IsoGSM systematically underestimates

the impact of convective events on the $\{q, \delta D\}$-pair distribution of a cloud-free troposphere, which in turn suggests that in our OSSE study we might underestimate the real remote impact of $\delta$D observations on convective events. For achieving the optimal benefit from the real world $\delta$D observations via a data assimilation approach, improving the modelled linkage between convective processes and the free tropospheric $\{q, \delta D\}$-pair distribution might be an important next step. In this context, the ongoing development of including water isotopologue simulations into different high resolution models also used

for operational weather forecasting (e.g. Pfahl et al., 2012; Eckstein et al., 2018; Tanoue et al., 2023) is very encouraging. A higher horizontal resolution and a convection permitting model setup (instead of parametrising convection as in IsoGSM) might further improve the capability of a model for correctly capturing the real world multi-scale impact of convective events (e.g. Pante and Knippertz, 2019) and thus better capture many details of convective processes (including the simulation of super-Rayleigh distributions).

## 5  Summary

We evaluate in detail the quality of the analyses of low latitudinal free tropospheric specific humidity ($q$), temperature ($T$), and water vapour isotopologue ratio ($\delta$D), as well as of the three water cycle variables free tropospheric vertical velocity ($\omega$), free

tropospheric latent heating rate ($Q_2$), and precipitation rate (Prcp). We investigate the impact of assimilating free tropospheric specific humidity and temperature (which can be easily observed by many different techniques) and the possibility of further improving the analyses by additionally assimilating free tropospheric water isotopologue data ($\delta$D), for which nowadays also reliable observations with good horizontal and temporal coverage exist. We assume that the observations are only available for cloud-free conditions.

First, we make a statistical overall evaluation considering all locations and time steps. The assimilation of $q$ and $T$ observations strongly improves the analyses data quality of $q$ and $T$ with skill values of up to 60%, if compared to no data assimilation. Concerning the analyses of the other variables ($\delta$D, $\omega$, $Q_2$, and Prcp) we also achieve a strong improvement with skill values of 10%-30%. Assimilating $\delta$D on top of $q$ and $T$ strongly improves the analyses of $\delta$D even further (leading to a skill value of up to 50%). However, the further improvement of the other variables ($q$, $T$, $\omega$, $Q_2$, and Prcp) is weak and we found that the overall impact of $\delta$D observations on the analyses quality is much smaller than the large impact caused by the observations of $q$ and $T$.

In a second evaluation we investigate how the $\delta$D observation impact depends on the atmospheric conditions. We use the atmospheric vertical velocity ($\omega$) as a proxy for atmospheric stability. The large majority of events represent stable conditions, for which the $\delta$D observations impact is generally negligible; however, for the rare convective conditions with strongly negative $\omega$, the $\delta$D observation impact is significant and for the analyses of the water cycle variables ($\omega$, $Q_2$, and Prcp) even slightly stronger than the $T$ observation impact. Although being rare, the very unstable conditions dominate the total yearly-averaged precipitation amounts in many regions and they are also related to extreme events (e.g. storms, flooding) that are not well captured in the analyses (for these extreme events also the analyses errors of $\omega$, $Q_2$, and Prcp are very large). This means that the $\delta$D observations offer potential for better capturing the events with the largest societal impact. Since unstable atmospheric conditions are almost always associated with clouds, we assimilate no observations at the location and time of these conditions. This hints to a unique remote impact of elsewhere available $\delta$D observations on the analyses of convective events.

A super-Rayleigh $\{q, \delta D\}$-pair distribution means high humidity and at the same time strong HDO depletion, and this distribution is linked to convective processes. We think that the conservation of these signals of isotopic depletion outside of the convecting area (where it can be measured) is essential for the unique remote impact of $\delta$D observations on the analyses of convective processes. In this context, we interpret the regular observation of super-Rayleigh distributions in the MUSICA IASI data as a promising indication that $\delta$D can have the same remote impact in the real that it has in our OSSE study for the simulated world. A real world $\delta$D assimilation works best, if the used model correctly captures the HDO depletion of convection. The availability of a growing number of high resolution atmospheric isotopologue enabled models importantly supports further progress in this field.

*Data availability.* The nature data and the ensemble mean data of the different assimilation experiments used for this study are available at https://radar.kit.edu/radar/en/dataset/PJeqXmWlLYSGBkJJ?token=rejHyXlETzWLGeopwLNq. The MUSICA IASI water vapour isotopologue data set is available at https://doi.org/10.35097/415.

*Author contributions.* Kei Yoshimura developed the isotopologue assimilation framework. Kinya Toride made all the data assimilation experiments. Matthias Schneider developed the ideas for evaluating the analyses improvements achievable by adding $\delta$D observations and made the respective calculations, whereby he was supported by Kinya Toride and Farahnaz Khosrawi. Frank Hase, Benjamin Ertl, and Christopher J. Diekmann provided important contributions for the design of this study. All authors supported the generation of the final version of this manuscript.

*Competing interests.* At least one of the (co-)authors is a member of the editorial board of Atmospheric Measurement Techniques.

*Acknowledgements.* This research has benefit from funds of the Deutsche Forschungsgemeinschaft (provided for the project TEDDY, ID 416767181).

Important part of this work was performed on the supercomputer HoreKa funded by the Ministry of Science, Research and the Arts Baden-Württemberg and by the German Federal Ministry of Education and Research.

We acknowledge the support by the Deutsche Forschungsgemeinschaft and the Open Access Publishing Fund of the Karlsruhe Institute of Technology.

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
