# Peer review of "Assessing the potential of free tropospheric water vapour isotopologue satellite observations for improving the analyses of convective events"

_EGUsphere, 2023_

## Referee Comment (RC1)

**Review of Schneider et al**

November 29, 2023

This article describes the possible added value of assimilating water vapor isotopic observations from the IASI satellite instrument in addition to assimilating humidity and temperature observations. To do so, observing system simulation experiments are performed with synthetic IASI observations. The main result is that there is a significant improvement in the case of extreme rainfall, due to the property of the isotopic composition to deviate from its usual relationship with humidity in strong convective conditions.

I have already reviewed previous versions of this manuscript submitted elsewhere, and this version is significantly improved relative to the previous versions. In particular, the added value of this article relative to previous studies, argued in lines 73-87, is very convincing.

The article is overall well written and illustrated. I have several comments.

**1 Major comments**

- l 19 and discussion in the text on the added value of $\delta D$ during "strong latent heating events": are IASI observations of good quality or frequent during the strong latent heating events, that are probably associated with cloudy conditions? We the impact of clouds on the retrieval quality considered when creating the synthetic IASI dataset? Maybe a few words could be added on this in the methods section? And possibly discussion section?

- Section 4.3 and fig 6: I'm not sure the link with the previous sections is clearly explained. I thought about this link and this is how I understand it: most of the time, $\delta D$ and $q$ are correlated, so the added value of assimilating $q + \delta D$ relative to $q$ is small. But for strong latent heating events, $\delta D$ deviates from its usual relationship with $q$, so this is where the added value of assimilating $q + \delta D$ relative to $q$ is the largest. Is this what the reader is supposed to understand? If so, maybe this should be explained more clearly, rather than letting the reader elaborate his/her own conclusion. If I misunderstood, then clarify as well.

- The results from Fig 3 to 6 were stratified by Q2: at which altitude?). Is there any reason for choosing to stratify by Q2 rather than precipitation rate or by $\omega$ at 500hPa, which are variables that are more commonly used in the community to stratify observations? Would the results be the same if they were stratified by e.g. precipitation?

- I understand that $\delta D$ allows to identify "strong latent heating events". In analyses, OLR observations are routinely assimilated. They are cheap and with excellent spatio-temporal coverage. I expect that OLR observations are very relevant to identify "strong latent heating events". Do we expect any skill improvement when assimilating $\delta D$ in addition to $q$, $T$, OLR?

**2 Minor comments**

- l 19: "most important": be more specific: e.g. needed due to the low skill? Or important for societal implications?

- l 24: "heating or latent heat consumption" -> "heating/cooling", for simplicity and coherence with the previous line.

- l 25: "impacting on" -> "impacting"

- l 164: "but we do not ... variables." -> "but that are not assimilated ."

- l 181: "calculation data of continuous" -> "calculation, continuous"

- l 241: "with significant we mean..." -> Write a full sentence outside of the brackets: "By significant, we mean..."

- "as already achieved by" -> "relative to that achieved by"

- l 257-259: clarify that although it provides information, the skill improvement is small.

- l 291: "almost not": why almost not? Why not completely not? In absence of any assimilation, don't we expect no relationship at all?

- l 292: "this uncertainties" -> "the uncertainties"

- Fig 3: recall which altitude this is. Same fig 4 and 5.

- l 380: "under which... analyses" -> "where the impact on the analyses is largest."

- l 386: "here used model IsoGSM" -> "IsoGSM model used here"

- l 391: "different highly resolving models" -> "convection-permitting models"?

- l 390-397: I'm not sure I understand the point of this paragraph: what is expected to have the largest impact on the analyses: the assimilation of real IASI $\delta D$, or the increased resolution? And is there any link between these two sources of possible improvement? If so, clarify. Regarding the impact of resolution on analyses, I suspect that there is already an extensive body of literature on this, maybe some papers could be cited?

- l 410: I didn't understand this sentence. Replace the sentence between brackets by just "the skill is improved by less than 10%"?

---

## Author Comment (AC1)

This article describes the possible added value of assimilating water vapor isotopic observations from the IASI satellite instrument in addition to assimilating humidity and temperature observations. To do so, observing system simulation experiments are performed with synthetic IASI observations. The main result is that there is a significant improvement in the case of extreme rainfall, due to the property of the isotopic composition to deviate from its usual relationship with humidity in strong convective conditions.

I have already reviewed previous versions of this manuscript submitted elsewhere, and this version is significantly improved relative to the previous versions. In particular, the added value of this article relative to previous studies, argued in lines 73-87, is very convincing. The article is overall well written and illustrated. I have several comments.

Many thanks for your support and careful reviewing of this manuscript. We particularly appreciate your efforts in evaluating this work in comparison to another (previously submitted) manuscript that used a similar set of assimilation experiments and to other previous studies that address this topic.

1 Major comments

Based on your comments we elaborated a revised version with important clarifications.

l 19 and discussion in the text on the added value of delD during strong latent heating events: are IASI observations of good quality or frequent during the strong latent heating events, that are probably associated with cloudy conditions? Was the impact of clouds on the retrieval quality considered when creating the synthetic IASI dataset? Maybe a few words could be added on this in the methods section? And possibly discussion section?

Yes, we agree. For strong latent heating/convective events there are mid- and high-level clouds. Under these conditions, thermal nadir sensors, like IASI or TES, offer no high quality free tropospheric products. For the revised version, we have repeated all the assimilation experiments. Now Sect. 2 explains in detail that for our OSSE we assume that there are no observations available for a cloudy model atmosphere, i.e. for unstable atmospheric conditions in the model. We also included an additional figure for documenting this. This change of the assimilation experiments does not qualitatively change the results of our study; however, it has required a modification of all the figures showing the skill values.

Justification of synthetic observation availability as used in the version of the discussion phase: As aforementioned, in the revised version we assimilate only observations, if the model indicates a cloud-free atmosphere. Nevertheless, we would like to explain, why we have used a different setup for the manuscript version in the discussion phase. There we have linked the availability of an observation in our OSSE to the availability of a real IASI observation. With this setup the {H2O,delD} pairs used as observations are distributed above and below a typical tropical Rayleigh line. This is similar to the respective distribution of the real IASI observations (see attached Figure, panel a). Because {H2O,delD} pairs below the Rayleigh line contain information on convective processes, this setup might ensure that the observations we use in our OSSE are affected to a similar extent by convective processes as the real

IASI observations. In contrary, defining the availability of observations by the clouds as simulated by the model, we find that almost all the {H2O,delD} pairs of the assimilated data lie above the Rayleigh line (see attached Figure, panel b). This is significantly different from the distribution of the real IASI observation and indicates that the model very likely underestimates the impact of convective processes on the distribution of the {H2O,delD} pairs, which in turn very likely underestimates the real impact of the observations on the analyses of convective events.

[Figure]

**Figure:** Distribution of {q,delD}-pairs at about 600hPa as seen in different data sets. Shown are contour lines for the highest {q,delD}-pair data density (thick and thin lines show the areas containing 90% and 50% of all the data, respectively). Panel (a), green contour lines: data used as observations in the assimilation study presented in the version of the discussion phase. Panel (b), green contour lines: cloud-free data from the nature run, i.e. the data used as observations during the assimilation step in the revised assimilation study. Blue contour lines: actual IASI observation data (the same in both panels). The dotted grey line is a typical tropical Rayleigh line, assuming the following atmospheric condition over the ocean source location: T=25°C, RH=80%, and delD=-80‰.

Section 4.3 and Fig 6: I'm not sure the link with the previous sections is clearly explained. I thought about this link and this is how I understand it: most of the time, delD and q are correlated, so the added value of assimilating q + delD relative to q is small. But for strong latent heating events, delD deviates from its usual relationship with q, so this is where the added value of assimilating q + delD relative to q is the largest. Is this what the reader is supposed to understand? If so, maybe this should be explained more clearly, rather than letting the reader elaborate his/her own conclusion. If I misunderstood, then clarify as well.

In the version of the discussion phase, we used Fig. 6 for discussing that the {q,delD} pair distribution for which the strongest delD impact is achieved, is actually observed in the real world. This was meant to justify a bit the selection of our OSSE observations. We agree that his was not very clear and we think that now by clearly limiting the observations to events that are cloud-free in the model the respective discussion in Sect. 4 becomes much clearer.

There are no observational data coinciding in time and space with convective events (the strongest latent heating events). In the real world we have no direct observations of convective processes because then there are clouds; however, there are many observations under cloud-free conditions that are affected by convective processes (e.g. for IASI in Schneider et al., 2017; Diekmann et al., 2021a or for TES in Noone et al. 2012). This is indicated by many {q,delD}-pairs located in the super-Rayleigh domain, i.e. below the Rayleigh line. In the revised we separate Sect. "Discussion and outlook" into two subsections: "Simulation versus real world data" and "Outlook on assimilating real world δD observations".

In Subsection "Simulations versus real world data", we use the {q,delD}-pair distribution figure for arguing, that the model used in our study very likely underestimates the impact of a convective process on the {q,delD}-pair distribution of a cloud free atmosphere (i.e. on a region outside of a convective area). Or the other way round, this figure fits now better to our argumentation chain: it suggests that the model underestimates the impact of convective processes on the cloud-free atmospheric {q,delD}-pair distributions, and we conclude that the real world IASI observations might have an even stronger impact on the analyses of convective processes than estimated by our assimilation study.

Subsection "Outlook on assimilating real world δD observations" discusses the challenges and possibilities for assimilating real world δD observations made by upcoming new satellites and novel isotopologue models.

The results from Fig 3 to 6 were stratified by Q2: at which altitude?). Is there any reason for choosing to stratify by Q2 rather than precipitation rate or by ω at 500hPa, which are variables that are more commonly used in the community to stratify observations? Would the results be the same if they were stratified by e.g. precipitation?

For the revised paper, we follow the recommendations of the referee and stratify the results with respect to ω, in order to be in line with what is frequently used by the community. This modification required changes of Figs. 4-7 and also of the title of the manuscript. Actually free tropospheric vertical velocity (ω), heating rate (Q2), and precipitation are strongly correlated, so it makes no qualitative difference if we chose ω, Q2, or precipitation for stratifying the results in Section 4.

I understand that delD allows to identify strong latent heating events. In analyses, OLR observations are routinely assimilated. They are cheap and with excellent spatio-temporal coverage. I expect that OLR observations are very relevant to identify strong latent heating events. Do we expect any skill improvement when assimilating delD in addition to q, T, OLR?

Yes, we agree, assimilating OLR might improve the analyses of the strong latent heating and convective events. We did not investigate this. In order to avoid that different spatial and temporal coverages affect the results, all the used observations (q, T, and delD at about 550 hPa) have the same spatial and temporal coverage, all these observations are only available for cloud free conditions. This is different for OLR, which can provide additional information for strongly cloudy conditions. OLR could provide important information on the location of convective events; however, this would require using OLR observations also at locations and time steps where the atmosphere is extremely cloudy. Using OLR

observations with a spatial and temporal coverage that is perfectly complementary to the other observations (q, T, and delD) makes it difficult to understand whether an improvement in the analysis is due to the complementarity of the observed parameter or due to the complementary coverage. In this study we want to avoid this difficulty (see introduction, line 69ff).

Moreover, a starting point of our study is the fact that the diabatic heating rates or convective processes obtained from different current global reanalyses show significant inconsistencies (Chan and Nigam, 2009; Ling and Zhang, 2013), despite that fact that OLR could be assimilated.

2 Minor comments

L 19: most important: be more specific: e.g. needed due to the low skill? Or important for societal implications?

At the beginning of the same sentence, it says that these are the extreme conditions with high precipitation and that for these conditions the analyses are rather uncertain. In our opinion this explains why it is "most important" and no extra sentence is needed.

l 24: heating or latent heat consumption-> heating/cooling, for simplicity and coherence with the previous line.

Similar comment by referee 2. We apply their recommendation: "[…] where latent heat is released or consumed"

l 25: impacting on -> impacting

Ok.

l 164: but we do not ... variables. -> but that are not assimilated

Ok.

l 181: calculation data of continuous -> calculation, continuous

Ok, thanks!

l 241: with significant we mean... -> Write a full sentence outside of the brackets: By significant, we mean...

Ok.

as already achieved by -> relative to that achieved by

This comment does not appear in the revised manuscript anymore.

l 257-259: clarify that although it provides information, the skill improvement is small.

Important is the comparison relative to the other observations. This comparison can be made quantitatively in the revised version by using the assimilation of q, T and delD together as the reference experiment. In the revised manuscript, we now state at the beginning of Sect.4 that "The overview study

of the previous section reveals that the delD observation impact is overall weak and generally much weaker than the respective impacts of the q and T observations".

l 291: almost not: why almost not? Why not completely not? In absence of any assimilation, don't we expect no relationship at all?

Without assimilating any data the model at least can separate the inner tropics (upward motion) from the subtropics (downward motion), i.e. even without data assimilation the model does correctly predict some climatological signals.

l 292: this uncertainties -> the uncertainties

Ok, thanks!

Fig 3: recall which altitude this is. Same Fig 4 and 5.

Ok, right, thanks! In the figure captions of Figs. 4, 6, and 7 of the revised manuscript we clarify what altitudes are represented.

l 380: under which... analyses -> where the impact on the analyses is largest.

This comment does not appear in the revised manuscript anymore.

l 386: here used model IsoGSM -> IsoGSM model used here

Ok, thanks!

l 391: different highly resolving models -> convection-permitting models?

This is clearer in the revised manuscript. There, we first mention the promising development of implementing isotopes in novel highly-resolving operational weather models, and then specify that the high resolution together with a convection permitting setup might be in particular important for further improvements:

"For achieving the optimal benefit from the real world $\delta$D observations via a data assimilation approach, improving the modelled linkage between convective processes and the free tropospheric {q,$\delta$D}-pair distribution might be an important next step. In this context, the ongoing development of including water isotopologue simulations into different highly resolving models also used for operational weather forecasting (e.g. Pfahl et al. 2012; Eckstein et al. 2018; Tanoue et al. 2023) is very encouraging. A higher horizontal resolution and a convection permitting model setup (instead of parametrising convection as in IsoGSM) might further improve the capability of a model for correctly capturing the real world multi-scale impact of convective events (e.g. Pante and Knippertz 2019) and thus better capture many details of convective processes (including the simulation of super-Rayleigh distributions)."

l 390-397: I'm not sure I understand the point of this paragraph: what is expected to have the largest impact on the analyses: the assimilation of real IASI delD, or the increased resolution? And is there any link between these two sources of possible improvement? If so, clarify. Regarding the impact of resolution on analyses, I suspect that there is already an extensive body of literature on this, maybe some papers could be cited?

For the revised manuscript, we modified this paragraph and have worked on a clearer argumentation chain: (1) super-Rayleigh {q,delD}-pair distributions contain valuable information about convective processes, (2) these distributions are frequently observed in the real world, but (3) they are under-represented in the model data. Our conclusion of (1)-(3) is, that there is high potential for achieving a significant impact of the delD observations on the model representation of convective events by real world delD assimilation. But we need models that better capture the link between convective processes and the free tropospheric {q,delD}-pair distributions on larger scales (between areas of convection and remote cloud-free areas). We discuss that high-resolution convective permitting models are promising for solving this problem, because it has been shown that such models have an improved performance for capturing larger scale atmospheric characteristics. As reference we give the work of Pante and Knippertz (2019).

l 410: I didn't understand this sentence. Replace the sentence between brackets by just "the skill is improved by less than 10%"?

This statement does not appear in the revised manuscript anymore.

---

## Author Comment (AC2)

The manuscript evaluates the use of assimilating dD measured by the satellite IASI in addition to traditional meteorological variables like q and T (also measured by IASI) in an Observation System Simulation Experiment (OSSE) with the isotope-enabled climate model IsoGSM. In general, dD adds little but notable skill to the analysis. Since dD carries information about the phase change history of air masses, the assimilation of dD particularly improves the analysis in cases of strong condensation or evaporation (identified by high latent heating/cooling rates). These extreme events usually involve strong precipitation and are therefore societally relevant, but are poorly captured in most analyses. Therefore the improvement for these events is very promising.

The manuscript is well-written and the figures are appropriate. I appreciate that everything is nicely structured and carefully documented. It is easy to follow the authors' explanations throughout the manuscript. I have a few comments that can hopefully further improve the manuscript and/or clarify some aspects.

Many thanks to the referee for reading our manuscript and for their efforts in evaluating the design and presentation of our study. The comments help in deed to further improve and clarify the manuscript.

General comments

1) As far as I know IASI does not see through clouds. Were observations in cloudy conditions filtered out before the assimilation? I would expect that strong latent heating events are almost always associated with clouds. If IASI cannot measure dD in these cases, is the improvement of the analysis thanks to dD still realistic?

This is an important comment and it has also been made by referee 1. MUSICA IASI data are only available when there are no clouds or when IASI's field of view is only very weakly affected by clouds. In the revised manuscript, we fully consider this limitation and calculated all skill values again. This required changes of Figs. 2-4, 6, 7 (numeration as in the revised manuscript). Some quantitative adjustments result from this, while qualitatively the results remain similar. In order to document the kind of atmospheric states that are represented by the observable events, we add a new figure (Fig. 1). This figure shows how the cumulative distribution functions of the different atmospheric variables change, if we only look on cloud-free events. For instance, the cloud-free observations represent only events with latent cooling (or low latent heating) and vertical downward transport (or weak vertical upward transport). For more details on the involved modifications and for an explanation/justification of the setup used in the version of the discussion phase please see our reply to referee 1.

2) The latent heating rate is defined as the change of specific humidity in an air parcel (material derivative of q) times the latent heat of net condensation (equation 9). However, mixing processes can also lead to a change in specific humidity in air parcels and could

therefore bias the analysis. Is there a way to separate mixing from latent heating/cooling, e.g. by diagnosing Q2 directly in the model?

To our understanding, the changes of q by horizontal or vertical transport are considered in Eq. (9) by the two terms "v nabla q" and "ω ∂q/∂ω". Or is there a misunderstanding? In any case and in line with a recommendation of referee 1, in the revised manuscript we now stratify the results with respect to vertical velocity (ω) instead of latent heating (Q2).

3) I think it might be nice to show also the horizontal spatial distribution (i.e. maps) of the differences in the skills to see the regions where the assimilation does or does not work well. Or is it pretty uniform?

We agree that the spatial context of skill changes is an important aspect. We analyse only 40 days of data with a 6 hourly resolution, and performing skill calculation for each location independently is very uncertain, because there are only 40x4=160 data points available for calculating the RMSD values. For small RMSD, the respective skill uncertainty is then often larger than the obtained skill value. In our study, we use large data amounts when calculating skills, i.e. use analysis data from many different locations and time steps together. This assures that the obtained skill values are above their estimated uncertainties. For spatial patterns of RMSE values, we would also like to refer to the paper of Toride et al. (2021).

4) Is there a reason why you use daily mean values for the analyses? I would expect that especially the strong latent heating/cooling events are rather short-lived and the improvement could be better on shorter time scales, e.g. 6 hours.

IASI makes observations at approximately the same location every 12 hour. If we furthermore account for the lack of observations when there are cloudy conditions, the observation frequency of a certain geolocation is generally less than 12 hours. For this reason our study in the version of the discussion phase was made with daily mean analysis, i.e. we focus on the time scales that are representative for the observations.

On the other hand, your argument is valid and using an OSSE, it should be possible to reconstruct part of the daily cycle in the analyses even though the observation frequency is sparser. For this reason, in the revised manuscript we present the evaluation of 6 hourly data, which, however, causes no qualitative change of the results.

Specific comments

L24: Suggestion: „where latent heat is released or consumed"

Ok, thanks!

L25: impacting on > impacting

Ok.

L39: As far as I know, it should be D and H in the equation (instead of HD16O and H216O).

There is also H218O or HD18O, so we suggest to leave as is.

L50: There is also an isotope version of NICAM (Tanoue et al., 2023), which might be worth adding here.

Ok, we add this reference.

L53: clouds or precipitation involving processes > processes involving clouds or precipitation

Ok, thanks.

L83: such assimilation > such an assimilation

Thanks for the comment, we will double check with a native speaker.

L108: What is different in the 96 initializations? Later you write the initial conditions. What exactly is different in the initial conditions?

The initial conditions for 96 ensemble members are chosen from the nature run from 0000 UTC June 1, 2016 with a 6-h time step, i.e. we apply a time shift in the intitial conditions of one month. This makes the initial conditions practically independent of the nature data one month later, but similar climatological conditions remain (for more details see Toride et al, 2021). We will add here the reference to Toride et al. (2021)

L155+: Add/Explain somewhere what is a good skill and what is a bad skill? E.g. 100% means perfect, 0% means same bad as the reference.

Ok.

L157 do do > to do

Ok, thanks.

L162+: This has been said many times already. I think it could be removed here or somewhere else.

Ok, we shorten here.

L171: „Therefore…": I don't see how this sentence follows from the previous sentence.

Ok, we remove "therefore".

L179: … and 17 vertical levels?

Yes, the analysis output is provided at 17 vertical levels, but we do not think that this is important here.

L208: 2x between

Thanks.

L257: close the > close to

This sentence has been modified in the revised version.

L281: uncertainty > error

Ok.

L292: this uncertainties > these uncertainties

Thanks.

L303: Remove „an"

Thanks.

L304: independent on > independent of

Thanks for the comment, we will double check with a native speaker (colleague).

L306: quantitatively document > quantify?

Thanks.

L309: How did you define these bins? Why not the same 60 bins as before?

In Fig. 3 (revised Fig. 4) we show the distribution of RMSD values for many equidistant ω bins. This overview shows that RMSD values are often small at ω close to zero. For Fig. 5 (revised Fig. 6) we use less bins with large bins for ω close to zero. The reason is that for ω close to zero, the skills are small and if we used smaller bins the respective skill uncertainty would be much larger than the obtained skill value, i.e., it would not be informative.

L343: in particularly > in particular / particularly

Thanks.

L346: by the additional assimilating of > by additionally assimilating / by the additional assimilation of

Thanks.

L359: These subset > This subset

This sentence has been modified in the revised manuscript.

L392: I would add NICAM here again.

Ok.

Figure 1, caption: remove „skill" after (e). Is „only q" not also „only one type of observation" (like „only T" or „only dD")?

Thanks.

Reference

Tanoue, M., Yashiro, H., Takano, Y., Yoshimura, K., Kodama, C., & Satoh, M. (2023). Modeling Water Isotopes Using a Global Non-Hydrostatic Model With an Explicit Convection: Comparison With Gridded Data Sets and Site Observations. Journal of Geophysical Research: Atmospheres, 128(23), e2021JD036419.

---

## Author Response (AR1)

Dear Editor,

In response to the major comments/suggestions of the referees' we made the following major modifications of the manuscript:

(1) We present a new set of assimilation experiments: for the new experiments we only assimilate observations when the model indicates no mid- and high-level clouds. The data characteristics for these cloud-free conditions are discussed in an additional figure (Fig. 1). We evaluate the analyses on a six hourly time scale, instead of a daily time scale. This requires modifications of all figures.

(2) Section 3.2: instead of calculating the complementarity of additional T or $\delta$D observations (using q and/or {q,T} as the reference), we investigate in detail the observation impact of all three observation types q, T, and $\delta$D (using a single reference: {q,T,$\delta$D}). This allows quantitative comparisons between the observation impacts of the "traditional observations" (q and T) and the observations impact of the "new" $\delta$D observations. This in turn leads to clearer conclusion about the uniqueness of $\delta$D observations. This modification requires updates of Table 3, Figs. 3 and 6, and Subsection 3.2. has been renamed from "Complementarity of additional observations" to "Observation impacts of q, T, and $\delta$D"

(3) We investigate the dependency of the skill/observation impact on the vertical velocity instead of the latent heating rate as in the manuscript of the discussion phase, in order to make it more consistent to other studies. This required changes in Figs. 4-6 and also of the title of the manuscript: from "Assessing the potential of free tropospheric water vapour isotopologue satellite observations for improving the analyses of latent heating events" to "Assessing the potential of free tropospheric water vapour isotopologue satellite observations for improving the analyses of convective events"

(4) We separate Subsection 4.3 ("Discussion and outlook") into two subsections: "Simulations versus real world data" and "Outlook on assimilating real world $\delta$D observations". The first subsection discusses the differences between the observations used in the OSSE and real world observations. The second subsections discusses the resulting challenges and possibilities for a future real world $\delta$D data assimilation. We think that this modification of the argumentation chain allows for a better understanding of our conclusions.

We also considered the minor comments of the referees (see the respective replies).

Many thanks for your support and best regards,

Matthias Schneider

---

## Author Response (AR2)

Dear Editor,

we would like to thank you again for your strong support in reviewing this manuscript.

We considered all your corrections/comments together with the corrections/comments of Referee #2 for this final revision (Referee #1 had no further comments). All the modifications can be tracked in the uploaded "latexdiff"-pdf file. We would also like to thank the two Referees for their help.

Best regards